# How Have Researchers Acknowledged and Controlled for Academic Work Activity When Measuring Medical Students’ Internet Addiction? A Systematic Literature Review

**DOI:** 10.3390/ijerph18147681

**Published:** 2021-07-20

**Authors:** Ken Masters, Teresa Loda, Finja Tervooren, Anne Herrmann-Werner

**Affiliations:** 1College of Medicine and Health Sciences, Sultan Qaboos University, Al-Khoud 0123, Oman; 2Department for Psychosomatic Medicine and Psychotherapy, University of Tübingen, 72001 Tübingen, Germany; Teresa.Loda@med.uni-tuebingen.de (T.L.); finja.tervooren@posteo.de (F.T.); Anne.Herrmann-Werner@med.uni-tuebingen.de (A.H.-W.)

**Keywords:** internet addiction, medical students, medical education

## Abstract

Internationally, medical students’ Internet Addiction (IA) is widely studied. As medical students use the Internet extensively for work, we asked how researchers control for work-related Internet activity, and the extent to which this influences interpretations of “addiction” rates. A search of PubMed, CINAHL, Web of Science, Scopus, and Google Scholar was conducted on the search phrase of “medical students” and “internet addiction” in March 2020. In total, 98 studies met our criteria, 88 (90%) used Young’s Internet Addiction Test, and the studies’ IA rates ranged widely. Little note was taken of work-related activity, and, when discussed, had little to no impact on the interpretation of Internet “addiction”. Studies seldom accounted for work-related activities, researcher bias appears to influence their position, “usage” appears conflated with “addiction”, and correlations between “addiction” and negative behaviours are frequently confused with one-way causation. In spite of IA’s not being officially recognised, few researchers questioned its validity. While IA may exist among medical students, its measurement is flawed; given the use of the Internet as a crucial medical education tool, there is the risk that conscientious students will be labelled “addicted”, and poor academic performance may be attributed to this “addiction”.

## 1. Introduction

### 1.1. The Internet and Addiction

The Internet, with some 1.8 billion websites and 4.5 billion users [1,2], is a vast, multi-faceted tool, crucial to all modern human activity, including work, research, study, and entertainment. In 1995, an individual’s reliance on the Internet was satirically termed Internet Addiction Disorder (IAD) by psychiatrist Ivan K. Goldberg, and the description intentionally included farcical symptoms such as “fantasies or dreams about the Internet” and “voluntary or involuntary typing movements of the fingers” [3].

Since then, however, Internet Addiction (IA) has become widely and seriously discussed. For example, a simple Google search on the phrase delivers more than 5 million hits, there have been arguments and support to have IAD (or variations of it) officially recognised in the Diagnostic and Statistical Manual of Mental Disorders (DSM) [4,5,6,7], and several researchers believe that it is already there [8,9,10,11].

### 1.2. Measuring Internet Addiction

In spite of the fact that IAD is not in the DSM, it is internationally routinely measured. It is difficult to know exactly how many IAD scales and variations exist, but a 2014 study [12] found at least 45 such tools across 23 languages. From that study, the most commonly used scale was Kimberley Young’s Internet Addiction Test, frequently called Young’s IAT, either in its original 8-question format [13], or more recent 20-question format.

The questions are typically asked on a Likert Scale, answered with “Occasionally”, “Frequently”, “Often” or “Always”; examples are these taken from Taha et al. [14]:

How often do you:Stay online longer than intended.Neglect household chores to spend more time online.Form new relationships online with fellow Internet users.[Have] others complain about the amount of time you spend online.Snap, yell or act annoyed if someone interrupts you while you are online.Lose sleep due to late-night Internet use.Find yourself saying “just a few more minutes” when online.Choose to spend more time online than going out with others.

The sense is that the questions indicate behaviour that is in opposition to healthy or desired behaviour. There is a great appeal to using these scales: the questions are presented on simple Likert scales, and numerical data can easily be extracted and then matched against a pre-set scale to determine the percentage of people who fit a specific classification group (e.g., “None”, “Mild”, “Moderate” or “Severe”). There are, however, other issues to consider when dealing with medical students’ use of the Internet.

### 1.3. Medical Students and the Internet

Even before COVID-19, medical students have long relied on the Internet for a wide range of educational and academic work-related activities, including accessing and submitting materials through their institutional Learning Management System (LMS), literature searching, and broader research. For academic purposes, the Internet is a vast and sophisticated library and a necessary tool for medical students’ work.

This reliance on the Internet as a library for academic work, however, introduces an uncomfortable disruption into the smooth flow of those earlier “addiction” questions when posed to medical students in Internet Addiction studies. A previous opinion piece by two of the authors (K.M. and A.H.-W.) [15] raised the argument that, if we had replaced the word “online” or “Internet” with “studying” or “library” and had put them to a 20th Century medical student, we would have these:

How often do you:Stay [in the library] longer than intended.Neglect household chores to spend more time [studying or in the library].Form new relationships with fellow [library] users.[Have] others complain about the amount of time you spend [studying or in the library].Snap, yell or act annoyed if someone interrupts you while you are [studying].Lose sleep due to late-night [studying or library] use.Find yourself saying “just a few more minutes” when [studying or in the library]Choose to spend more time [studying or in the library] than going out with others.

It is unlikely that anyone would have done such research in the 20th Century, because the answers would be foregone conclusions. If these questions give an indication of “addiction”, then would medical students be considered “addicted” to the library?

Given medical students’ usage of the Internet, it is crucial to distinguish how much of that usage is for work-related activity, so that this work-related activity can be separated from non-productive and even damaging usage that would more traditionally be associated with addictive behaviour.

From this background, we derive our overall question: given medical students’ use of the Internet for work-related activities, how do IA studies control for this, and ensure that they are measuring addiction and not reliance on a tool necessary for study? While other systematic reviews of IA research exist (e.g., [16,17,18,19,20]), their focus is not always on medical students only, and they tend to ignore the issue of work-related activity, focussing instead on the IA statistics themselves. In order to better understand the reported level of medical students’ IA as measured by these tools, we need to view these levels when controls for work-related activities are taken into account.

In addition to determining the extent of studies’ measuring IA in students, in this review, we ask of these studies:Do they question the validity of IAD?Do they acknowledge that medical students use the Internet for academic work-related activities?Do they determine the extent to which their sample is using the Internet for academic work-related activity?If they do measure this activity, to what extent do those results affect the interpretation of the “addiction” rates as measured by their tools?

## 2. Materials and Methods

The Methods below follow recommendations from the Preferred Reporting Items for Systematic Reviews and Meta-Analyses (PRISMA) Guidelines [21]. The recommendations in the STructured apprOach to the Reporting In healthcare education of Evidence Synthesis (STORIES) [22], and Association for Medical Education in Europe (AMEE) Guide [23] were also consulted for appropriate guidelines.

A research protocol was created by KM, and all authors gave input, reached consensus on its contents. The authors divided into two teams (KM; AH-W, TL, FT) and independently conducted the literature searches during March 2020. Databases were PubMed (including Medline and PubMed Central), CINAHL, Web of Science, Scopus and Google Scholar (excluding patents) (See Figure 1). The search phrase was “medical students” AND “internet addiction” in the text. We realised that this would result in a wider initial range of papers than if we had sought these only as keywords, but we wished to capture as many studies as possible.

Included: Quantitative and qualitative, original, peer-reviewed, research papers (in journal or conference proceedings where the full paper was available) on the topic of internet addiction among medical students, in English. Papers in which the sample included non-medical students only if data for medical students were given separately. Post-publication, peer-reviewed papers only if the journal had a rating or distinction process, and the paper had met that rating. (Inclusion in an indexing service like Medline, PMC or Web of Science would be sufficient to qualify). There was no restriction on publication date or location of research.Excluded: Letters, opinion papers, commentaries, theses, and literature reviews.

After the initial search, the teams combined, and, through consensus made alterations. See Figure 1 for details.

Variables to be extracted were: year of study, sample country, sample size, age and gender, and brief description, study method, tools or methods used to measure addiction, whether or not the validity of Internet Addiction as a concept was questioned, whether or not the results were compared to (or measured against) other measurements (e.g., depression), addiction rates, description of any acknowledgment of the value of the Internet for academic work, other pertinent results and inferences (e.g., correlations) drawn.

When considering “academic work,” we looked for indications of specific activities (such as research, assignment submission, accessing the LMS) or use of the terms “academic work”, “education” or similar.

We looked for the possibility of bias primarily in the attitude towards the Internet in general, and the Internet as a work tool. In addition, any discussion around validity of Internet Addiction as a concept would indicate the level of objectivity or bias.

The principal summary measures were the variables stated above. In many cases not all the data would be explicitly stated, but could be calculated from the other data given. When this was done, this would be clearly reported in the results tables.

The data were extracted and tabulated independently by the two teams, and then resolved through consensus and re-checked by the lead author.

## 3. Results

Figure 1 shows the search process.

In total, 98 studies in 102 publications met our criteria and were included in the study. Appendix A [6,7,8,9,10,11,24,25,26,27,28,29,30,31,32,33,34,35,36,37,38,39,40,41,42,43,44,45,46,47,48,49,50,51,52,53,54,55,56,57,58,59,60,61,62,63,64,65,66,67,68,69,70,71,72,73,74,75,76,77,78,79,80,81,82,83,84,85,86,87,88,89,90,91,92,93,94,95,96,97,98,99,100,101,102,103,104,105,106,107,108,109,110,111,112,113,114,115,116,117,118,119,120] lists the studies and gives a brief description of each, and Appendix A [6,7,8,9,10,11,24,25,26,27,28,29,30,31,32,33,34,35,36,37,38,39,40,41,42,43,44,45,46,47,48,49,50,51,52,53,54,55,56,57,58,59,60,61,62,63,64,65,66,67,68,69,70,71,72,73,74,75,76,77,78,79,80,81,82,83,84,85,86,87,88,89,90,91,92,93,94,95,96,97,98,99,100,101,102,103,104,105,106,107,108,109,110,111,112,113,114,115,116,117,118,119,120] gives more detailed results, including the addiction rates as measured by the researchers.

Of these studies, 21 (21.43%) had no date of study, 2 (2.04%) were completed in 2009, 5 (5.10%) in 2012, 8 (8.16%) in 2013, 3 (3.06%) in 2014, 20 (20.41%) in 2015, 10 (10.20%) in 2016, 9 (9.18%) in 2017, and 20 (20.41%) in 2018.

There was a strong dominance of studies from one area of the world, and this is shown graphically in Figure 2.

Given that the Internet and medical students are both globally represented, one would assume that a systematic review of articles combining IA in medical students would be globally represented. In fact, given that much of the English-speaking Internet is American and that Kimberly Young is based in the USA, one would expect a bias of papers from the USA. This is not the case, however, and just why this should occur, and what impact it would have on the results would be an interesting area of further research.

A total of 36,397 students (mean = 37.39, median = 245.5) were involved. The smallest study had 83 students, the largest had 3738.

Of these studies, 88 (89.80%) used Young’s IAT (or variations of it), and three (3.06%) used Chen’s IAS (or variations of it); 92 (93.88%) did not question the validity of IA (of which three stated incorrectly that it is in the DSM, 12 acknowledged that it is not in the DSM, one acknowledged that there is no consensus on the definition, and six acknowledged work was still ongoing).

A total of 47 studies (47.96%) used an IA scale only, and the other 51 (52.05%) used an IA scale in combination with other measures.

On the question of academic work, 55 (56.12%) acknowledged (usually only in the Introduction) that the Internet is useful for work, but did not raise this idea in their Discussion, or they dismissed it as irrelevant to their findings, in spite of the fact that 39 (39.80%) found that their students used the Internet for academic work purposes.

Several of the papers (e.g., [9,10,33,70,84,115]) found associations between IA and negative behaviour, including poor academic performance, leading the authors to conclude that IA impacts negatively on students, and also causes poor academic performance. In addition, 25 of the studies mention that the term IAD was first proposed by Ivan K. Goldberg, but only three [73,84,89] note that it was introduced satirically.

## 4. Discussion

This systematic review has examined 98 studies dealing with Internet Addiction among medical students. The large number of studies included in the review indicates that IA in medical students has been extensively studied. Although the levels of IA as measured in the research have been reported, the prime goal was to assess the extent to which the studies acknowledged and controlled for the fact that medical students use the Internet for work-related activities. Overall, we have found that, while many studies briefly acknowledge the value of the Internet for academic work-related activity, most do not; in addition, most do not measure its impact on academic work, and, when they do, they largely ignore their own findings, and do not control for it when measuring IA.

In particular, however, there are other findings upon which we need to reflect.

### 4.1. Researcher Bias

A strong element in the studies is researcher bias, and it appears in many forms. Not all studies display all these biases, and they are probably unintentional, but they are frequently found. Broadly, there are four bias indicators.

#### 4.1.1. Bias Indicator 1: Ignoring the Value of the Internet in General

The necessity of using the Internet in almost every aspect of modern life and the value derived from it is either viewed in a negative light or ignored. For example, an opening statement like “The extensive use of internet in the last decade has become a major concern” [73] sets the context in which Internet usage is inherently problematic. Patel et al. [86], after noting that “students used to play outside for entertainment” state that a “worst situation is created with easy to access internet facility and social networking.”

Similarly, researchers do not consider the possibility that what is perceived as “addiction” is merely “usage.” So, when Ghosh and Chatterjee [56] find that over 70% of the students use the Internet for shopping and 80% for reading news, this does not lead them to the consideration that their students use the Internet for daily and necessary activities, and that these activities would influence the “addiction” scores.

This attitude is not unique to these studies, as there are some calls for laws to govern the use of Internet, especially among the youth [121].

#### 4.1.2. Bias Indicator 2: Ignoring the Value of the Internet to Work

Given that all of these studies were performed among medical students who would surely be using the Internet for academic purposes, it is noteworthy that many studies do not acknowledge its value, and, if they do, do not measure its use for work. When it is measured, it is then often ignored in the Discussion and not taken into account when interpreting the “addiction”, even when their results indicate high usage for work-related activities (e.g., [31,36,39,40,41] and many others). Chaudhuri et al. [44] recommend that Internet access be curtailed at night and in classes, in spite of finding that 83% of their population use it for work-related activities; Singh et al. [106] find a correlation between IA and Internet usage in the classroom and deem this to be problematic, and do not raise the possibility that this usage may be to look up educational material while in class.

The researchers’ reactions to different student levels’ affecting IA measurements also demonstrates bias: they do not question the higher “addiction” rates in their PhD students [59] or final year students [77], and Wang et al. [117] explain the increased addiction in the senior medical students is perhaps because “they have more time and energy”.

In addition, when measuring activities, separating social media and email activity from educational activity makes little sense unless one applies the bias that neither social media nor email can be used for educational activities.

Where studies report a low Internet usage for academic purposes, the researchers do not question their results or find this remarkable. Given that the electronic distribution of learning materials is wide-spread at medical schools (usually through an LMS), it is indeed strange that the figures report low usage of the Internet for work-related purposes. The researchers do not question how these students are accessing their learning materials, performing research or submitting assignments. (There may be cases where the location is so under-resourced that the Internet is barely functional; in that case, then, there would be little chance of addiction.)

#### 4.1.3. Bias Indicator 3: Not Questioning the Validity of IAD

In spite of the fact that IAD is not clinically recognised, this weakness is mostly ignored, even by those who acknowledge that it is not in the DSM. In almost all cases where it is noted, it is merely mentioned and then ignored. It appears that IAD is deemed axiomatic, and it can then be correlated with other negative traits measured by other instruments.

Perhaps part of the problem lies in the fact that, while 25 of the studies noted that psychiatrist Ivan K Goldberg first introduced the term IAD, the researchers appear unaware that it was satirical, and not meant to be taken seriously.

#### 4.1.4. Bias Indicator 4: Confusing Correlation with Causation

In several cases, correlation between Internet usage and negative traits are stated explicitly as causative [9,10,33,70,84,115]; in others they are implied. For example, Suresh et al. [110] write “As the level of internet addiction increased, i.e., from mild to moderate, subjective happiness decreased.”

This causative view ignores the fact that many struggling students are using the Internet to find useful information. This is an unfortunate and dangerous position to hold, because, instead of alerting educators to the fact that their struggling students are resorting to Internet sources to understand their material, the educators will be encouraged to believe that the poor performances are *caused by* their students’ Internet Addiction. This erroneous belief may lead to further calls for greater Internet-access restrictions, further damaging these students.

Correspondingly, correlations between IA and positive traits are dismissed. For example, Alpaslan et al. [30] find a high correlation between IA and grades, but dismiss their own results with the comment that they are “probably related to self-report of the students”—it is noteworthy that none of their other results are dismissed because of the self-reporting. Similarly, the lower failure rates among the “addicted” students does not impact on the interpretations of addiction by Komleh at al. [71].

In addition, many of the studies show correlations between IA and common online activities such as emailing, blogging, etc. Again, the researchers fail to mention the obvious: the tool measuring IA is not necessarily measuring addiction; it is measuring usage, and so having further questions about activities is bound to show a correlation, because those activities are merely the details of the usage. In essence, the researchers are finding that the students who use the Internet a lot, use the Internet a lot. It is hardly surprising that that there is a strong correlation between the “addiction” measurement and duration of usage: they are frequently one and the same thing.

### 4.2. Location

The location bias has been noted in the Results. As there is no clear indication of why this bias should occur, this opens a new area for research.

### 4.3. Range of Different Rates of Addiction

The wide range of addiction rates (Appendix A) is striking, and has been noted by other researchers [122]. One advantage of the wide range is that it reduces the risk that this review has been influenced by publication bias. The disadvantage, however, is that it is difficult to create a consistent global picture. Given the wide variation, even within regions and countries, this may also raise questions about the validity of the tools used.

### 4.4. Changes in the Air

From the discussion above, we can see that there are problems with the studies of medical students’ IA, but there is some suggestion of changes in the air.

Firstly, Siraj et al. [107] lists numerous studies showing the beneficial value of the Internet to academic work by medical students, and their own study demonstrates this benefit, although they appear to somewhat reluctantly acknowledge this: “Although there has been growing concern regarding the risk of overuse of internet, we cannot prevent its usage as there are many benefits linked with it.” Although they do not question the IA tool, their paper appears to end with an indication that perhaps all is not as it seems: “Further study is suggested with wider instruments for in depth study in order to investigate the dependent users…”. Unfortunately, this final sentence ends rather enigmatically with “…and also to take measures to rehabilitate them if necessary.” The use of “rehabilitate” implies that the dependence is possibly a disorder, and not because of work.

Secondly, other researchers [120] have called the IA measuring tools into question, and have suggested alternatives.

Thirdly, in early 2020, the IAT was revised [123], and is now called the Internet Addiction Test-Revised (IAT-R). While several new questions appear to take into account a work-non-work division, there are still many generic questions such as:Do you find that you use Internet activities longer than you intended?Do you neglect self-management (e.g., cleaning and eating) to spend more time on Internet activities?Do you form close relationships with other online users?Do family members or friends complain to you about the amount of time you spend on Internet activities?Do you lose sleep due to late night Internet activities?Do you try to cut down the amount of time you spend on Internet activities but fail?

These questions, asked about library usage to students 30 years would still have indicated an “addiction” to the library, when, in fact, they might more accurately describe deeply motivated students. So, while changes to the IAT are welcomed, these do not go nearly far enough. In fact, there is a danger that the new scale will give a new generation of researchers the false impression that the problem has been solved, and that the tool very accurately measures IA.

It is apparent then, that, while there may be a need for a tool to measure IA, it should be constructed by a multi-disciplinary team that includes educationalists, technologists, psychologists and others, which takes into account the specific needs of students. In fact, given that professional needs will differ across groups of students, it may be necessary to have different tools (or variations of one tool) designed for different professional areas. Further research and testing will allow for a clearer understanding of the problem.

### 4.5. IA and COVID-19

It would be unfair to judge the papers in the light of COVID-19, but events in 2020 do serve to give us pause, primarily because of a global shift towards online education. It is not that medical students are now using the Internet for work for the first time, but that their usage has increased significantly, as they now use it for accessing materials, classes, examinations and all manner of academic work. This significant increase serves to illustrate beyond doubt the inherent weakness of the IA tools, because IA research performed in 2020 would invariably lead one to believe that the majority of medical students in world are suddenly addicted to the Internet. This is clearly a fallacy, and so points us to the inevitable conclusion that any attempt to measure IA has to control completely for work-related activities. It is now the turn of the researchers to determine how best to do that.

### 4.6. Does IA Exist At All?

In this paper, we are not denying the existence of IAD (indeed, some studies (e.g., [114]) did find troubling relationships between IA measurements and work activities). We are saying, however, that the tools that currently intend to measure IAD do not measure it. They may measure reliance on the Internet in our daily lives, and, so, they may measure the students’ reliance on the Internet for many things, including work. In addition, given the prevalence of medical student burnout, they may be measuring medical students’ “addiction” to their work, in much the same way that previous generations’ reliance on the library may have done.

### 4.7. Limitations

Perhaps the prime limitation of this research lies in the location of the studies; as they were limited to particular regions of the world, similar results, and similar interpretations of results, may not be found elsewhere. In addition, although most of the studies used the same measurement tool, they did not always used the same sub-category classifications on the levels of the addiction. As this research focused more upon the researchers’ attitudes towards the addition in a work-related environment, the impact of this variation is not as great as it might be.

## 5. Conclusions

This review has examined the results of studies purporting to measure Internet Addiction in medical students. A wide range of results from numerous studies has been found, but the central weaknesses appear to be inherent biases of the researchers, especially ignoring the students’ work-related activities, and so using tools that do not measure what they set out to measure.

In direct response to our research questions, we have found that, in general: the researchers do not question the validity of IAD (even when acknowledging that it is not a recognised disorder); while more than half briefly acknowledge that medical students use the Internet for work-related activities, less than half measure it in their own research; when they do measure it, they almost always ignore it in their Discussions. The answer to our question in this paper’s title, then, is that researchers do not control for academic work activity when measuring medical students’ Internet Addiction.

While medical students’ Internet Addiction may be a problem, we conclude that the tools used by the researchers (and even the IAT-R) are far too blunt, and that they need to be far more granular, and specifically control for work-related activities. In this light, we recommend that a multi-disciplinary team create a revised tool that can accurately measure Internet Addiction in students. Unless this happens, the inaccurate measuring will have negative effects on medical education—not least of all is the risk that hard-working medical students will be labelled as “addictive”—and of those students, if their grades are not exemplary, then this “addiction” will be perceived as a cause of poor performance. Until this change is made, the results of IAD studies among medical students should be treated with great caution and possibly even ignored.

## Figures and Tables

**Figure 1 ijerph-18-07681-f001:**
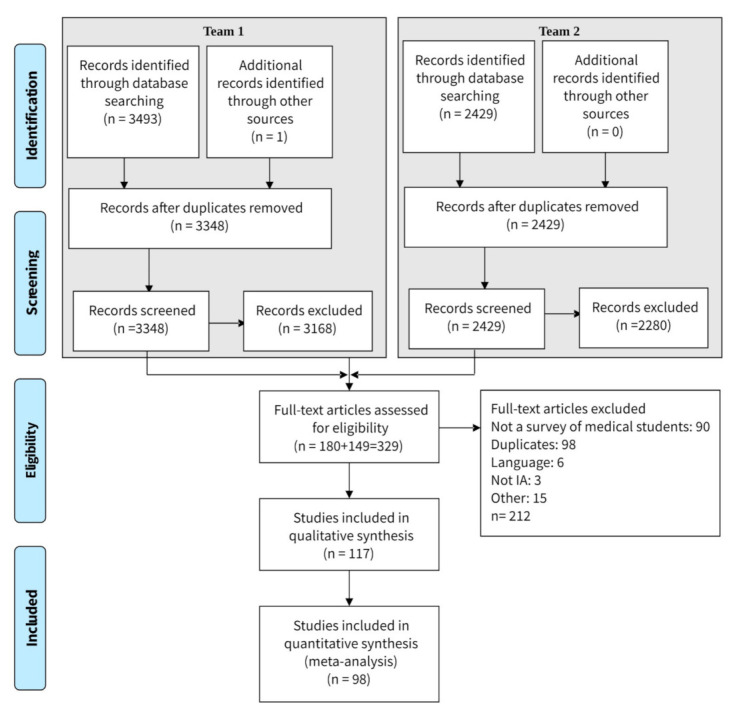
Flow chart of search details (Following PRISMA, 2009 [21]).

**Figure 2 ijerph-18-07681-f002:**
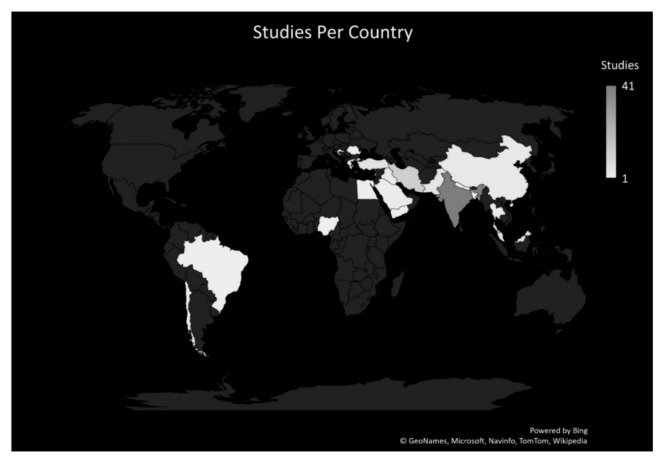
Geographical Location of Studies.

## Data Availability

Not applicable.

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
