# Peer review of "How Have Researchers Acknowledged and Controlled for Academic Work Activity When Measuring Medical Students’ Internet Addiction? A Systematic Literature Review"

_ijerph, 2021, doi:10.3390/ijerph18147681_

Round 1

Reviewer 1 Report

The article presents a systematic review of the literature based on Internet use by medical students. The title is "How have researchers acknowledged and controlled for aca-2 demic work activity when measuring medical students' Inter-3 net Addiction? A systematic literature review".

To provide an overview of the area of research related to internet addiction, this paper reports the results of the systematic analysis of relevant articles published up to early 2020.

I enjoyed reading your article, and it is relevant to the mission of the journal, this means that the topic is well within the scope of the journal. The topic of the article is interesting and a timely study, fundamentally in times of healthcare crisis where internet use has increased according to all studies. There is a need for research that facilitates understanding and interest in the problems of internet addiction, there is an increasing amount of scientific research literature on this topic.

The document is well structured, facilitating the understanding of the study conducted.  It also has a good theoretical introduction, where the concepts under study are clearly defined.

Objective: The research problem and the objective of the study are well defined.

Method: the study presents a systematic analysis of the literature using the PRISMA methodology. The research phases are presented in a clear and structured manner.

Results: this reviewer considers that the results shown in relation to the study problem are relevant and lead to clear conclusions. The figures and tables presented contribute to facilitate the reader's understanding.

Although the conclusions are clearly specified, the research questions should be made explicit to facilitate the reader's understanding. That is to say, to give answers to the questions raised in the study in an organized manner.

Specify also what have the authors learned from the findings that can guide the educational community to improvements in this field? Such a discussion would considerably strengthen the contribution.

In general, I consider it a good work that contributes effectively to the advancement of the field of knowledge.

Reviewer 2 Report

I really appreciate the opportunity to review this manuscript entitled “How have researchers acknowledged and controlled for academic work activity when measuring medical students’ Internet Addiction? A systematic literature review”. This is important to assess Internet Addiction (IA) in medical students and deal with the use they do.  I only remark some issues (most of them in methods) in order to improve the quality of this manuscript.

The abstract is clear but it is important to explain the last idea (line 22), why poor academic performance may be attributed to this “addiction”? Introduction was well structure and shows the necessity, and also the difficulty, for this research. The aim of the paper is clear and I think it should be justify, why it is important to assess Internet Addiction related to work activities?

At the methods section, there are some questions that should be review. About the inclusion criteria, authors do not remark temporal limits, were they taken into account? If they did take into account, they should include them.

Results were clear, may be some hypothesis looking for reason of non-papers published about this topic in United State should be developed in the Discussion. I have no access to supplementary fields to see the tables 1 and 2.

Discussion summarize and explain in a good way the finding but, from my point of view it would be interesting to discuss (line 321-322 ) about the importance of looking for relations among IA and work activities, and so on to suggest some tool to assess it.  On the other hand nowadays there is a great concern about IA? With these result, what could be accomplish?

Conclusions were correct, but I think again, that line 363-364 do not emerge from your results.

Reviewer 3 Report

The paper aims to analyze the literature on AI with a critical review aimed at understanding whether the methodology adopted to measure the phenomenon is correct or not. The consequences of a misunderstanding of the results obtained from various researches are clearly explained. A discussion is offered on the causes, but also the on the consequences, of this misjudgment. The paper therefore, while not constituting an advance in the knowledge of the phenomenon, nor in the methodological setting, is very useful for overcoming the superficiality with which it was faced in the past, especially in a historical moment that has increased the use of resources on a global level of the network for obvious health reasons.
The selection of the papers submitted for evaluation presents geographic limitations, really mentioned by the authors. However, the number of papers analyzed is sufficient to demonstrate the authors' theses and to reveal a real problem on this important topic.

Reviewer 4 Report

The article it may be of interest to researchers studying the subject.

It meets all the formal parameters for publication.

Conceptualization, methodology, validation, firm analysis, resources and bibliographic references, etc... They are suitable.
